# New Adverse Drug Reaction Signals from 2017 to 2021—Genuine Alerts or False Alarms?

**DOI:** 10.3390/pharmacy12010033

**Published:** 2024-02-10

**Authors:** Yoon Kong Loke, Katharina Mattishent, Navena Navaneetharaja

**Affiliations:** Norwich Medical School, University of East Anglia, Norwich NR4 7TJ, UK

**Keywords:** adverse events, disproportionality analysis, pharmacovigilance

## Abstract

Spontaneous adverse events reporting systems are used internationally to flag new or unexpected adverse drug reactions (ADRs). Disproportionality analysis is a recognised technique, but false alarms may arise. We aimed to determine whether these new ADR signals had subsequently been followed-up with detailed hypothesis-testing studies. We searched PubMed to identify published studies (years 2017–2021) where the authors reported findings of new ADR signals from disproportionality analyses. We used PubMed and forward citation tracking (Google Scholar) to identify any subsequent confirmatory studies of these ADR signals. We screened 414 titles and abstracts and checked the full-text articles of 57 studies. We found signals for 56 suspected new ADRs from 24 drugs. Google Scholar showed that the ADR studies had been cited a median of seven times (range 0–61). However, none of the suspected new ADRs had undergone detailed evaluation in the citing literature. Similarly, our PubMed search did not find any confirmation studies for the 56 suspected new ADRs. Although many suspected new ADR signals have been identified through disproportionality analysis, most signals have not been further verified as being either genuine ADRs or false alarms. Researchers must focus on follow-up studies for these new signals.

## 1. Introduction

Data on adverse drug reactions (ADRs) have a critical role to play in helping regulators, healthcare professionals and patients make informed decisions on the benefit-to-harm ratio of a treatment. However, the diversity and huge range of ADRs pose major analytical challenges [1]. Researchers and clinicians who hope to build a full picture of a medicine’s safety must carry out a comprehensive evaluation across a wide spectrum of data sources. Typically, evidence from randomized trials is supplemented by non-randomized studies that may aim to record data in real-world clinical practice [2]. We recognize that new medicines can potentially trigger rare, unexpected or previously unknown adverse reactions, and so there is a need for ongoing monitoring to detect serious adverse events that could come out of the blue. Clinicians and patients are able to take part in a voluntary system and submit reports of adverse events involving publicly available medicines for further assessment by regulatory authorities [3]. These post-licensing studies can detect rare events because they potentially have a far larger sample size, broader population coverage and longer durations of follow-up than would be available in early randomized trials.

Spontaneous adverse events reporting systems have therefore been set up internationally to record the whole spectrum of potential harm caused by medications [4]. For instance, the World Health Organization collates reports from 150 member countries across the world in its database (VigiBase). A major role of spontaneous reporting systems is to facilitate the early or rapid detection of newly emerging and/or unexpected signals [1]. Recent advances in these reporting systems have transformed what was once a medically dominated process to a wider system that is now open to reporting by patients and other healthcare professionals (such as pharmacists and nurses). This has significantly broadened the recording of adverse events and goes some way towards addressing the criticisms of the underreporting that is well recognized within spontaneous reporting systems, where only a small proportion of clinical adverse events are actually submitted to the relevant regulatory authority [4]. Here, it is hoped that the sensitivity of the system can be improved by enabling broader, more complete detection and by increasing the numbers of submitted reports. The overarching presumption here is that fewer potentially serious but rare adverse events would be missed if the reporting system has a sufficiently comprehensive coverage of the population exposed to the drug.

Some of the strengths of spontaneous reporting systems also happen to be major limitations. Broad detection, with provisions for patients and other healthcare professionals to make spontaneous reports, is essential. However, this also means that the completeness and the amount of detail within the reports can be highly variable [2]. Members of the public may use different terms to report their symptoms, and they may not be able to arrive at or submit accurate medical diagnoses in such reports. Equally, access to a patient’s full healthcare record may not be possible and therefore a spontaneous report might be based on incomplete data records of the patient’s medical conditions and the associated treatments. For instance, capturing the recorded comorbidities as well as the dosage and timing of potentially interacting medications can be a major challenge if the person submitting the report does not have full access to the complete healthcare record.

This overarching aim of capturing new, previously unrecognized or unexpected adverse events also relies on reporters submitting all types of adverse outcomes, irrespective of whether they think there is a causal relationship or not. Clearly, a spontaneous reporting system would fail to detect new signals if patients and clinicians only reported events that were already well recognized as adverse drug reactions. This broad ascertainment, however, means that many reported adverse events may not be specifically related or even have any direct link to the drug [2]. For instance, patients may experience adverse events that are related to the disease or to some other external variable (e.g., air pollution, food) that has no link to their medical treatment. The database is therefore possibly inundated with reports that may not be relevant, or may not be associated with the specific drug of interest [1]. Identifying a relevant signal is therefore something akin to looking for a needle within a haystack.

The above-mentioned limitations create major challenges to the interpretation and analysis of spontaneous adverse reports. In many ways, broad, diverse monitoring works very well when focusing on increasing the sensitivity of signal detection, but this may come at the cost of poor specificity if the system becomes swamped with reports that have no causal relationship or which lack sufficient detail for comprehensive assessment. It is beyond the scope of this article to cover all the techniques that have been suggested, but in the following paragraphs, we will discuss the key issues that need to be overcome when attempting to identify new ADR signals.

Reports of spontaneous adverse events can be evaluated using a variety of different approaches [2]. A qualitative or semi-qualitative approach can be taken through detailed clinical review of each individual report. This typically involves making some sort of judgment on the likelihood of the causal relationship between the suspected drug and the adverse event, which may involve semi-quantitative methods or causality algorithms. However, these qualitative assessments are directed at the level of an individual patient to determine likelihood of their particular drug treatment being causally responsible for an adverse event. In contrast, quantitative methods such as disproportionality analysis have been widely deployed to statistically assess whether a signal of harm can be found within large numbers of spontaneous adverse event reports in the wider population. The focus of disproportionality analysis is to determine the instances where there is a potentially large difference between the proportions of adverse events reported with a particular medication as compared to the proportions with another medication (or the proportion reported overall). Key features and major limitations of disproportionality analysis are listed in Table 1 [1,2,5].

Previously, signal detection using these quantitative methods was mainly driven by regulatory authorities and pharmaceutical companies who collect and analyse these spontaneous reports, with subsequent regulatory updates of the product information regarding any confirmed or validated ADRs [6]. However, wider public access to such databases has now enabled clinicians and researchers to conduct and publish results of signal detection studies outside of the regulators and pharmaceutical industry. The importance of these studies (conducted outside the purview of the regulatory framework) has recently been questioned [6].

Two meta-epidemiological assessments (covering 100 published studies involving disproportionality analysis) have identified major weaknesses in the methods and conclusions of the studies [7,8]. In the researchers’ opinion, the inconsistent methodology and lack of transparency in the signal-generating studies raise major concerns that such analyses could produce ‘misleading results and generate unjustified alarms’ [7]. This research team also judged that “40% used causal language to interpret their results in the abstract or conclusion” [8]. Over-interpretation of signal-generating data, coupled with failure to appreciate methodological limitations, could create potentially misleading impressions regarding causal relationships between drug use and adverse events [8].

Other researchers have also attempted to validate the results of disproportionality analyses through comparison with randomised trials or observational studies. Beau-Ljedstrom et al. conducted an evaluation of adverse events that had been selected randomly and reported only a weak correlation between safety signals arising from disproportionality analysis as compared to the findings of Cochrane systematic reviews that looked at those specific adverse events [9]. Conversely, another team of researchers set out by defining a known set of 15 adverse reactions and reported that disproportionality analysis yielded comparable findings to that of the observational studies [10].

More recently, in a letter published in the *British Medical Journal*, Khouri et al. asserted that there had been an exponential increase in the number of signal generation studies, but most of these signals had failed to be noticed or resolved [11]. In order to empirically assess the validity of Khouri’s assertion, we aimed to determine whether new signals arising from disproportionality analysis had undergone further validation and confirmed or refuted these signals using more formal methods. Our current objective and methodology also mirrors that of our previously published work regarding the lack of subsequent verification of validation of published case reports of adverse drug reactions [12].

## 2. Materials and Methods

We constructed a cohort of published reports (dates 2017–2021) of new suspected ADRs identified through disproportionality analyses. This was performed through a database search and screening of eligible articles using the following methods.

### 2.1. Search Strategy

We searched Pubmed on 8 June 2023 with the following search terms: adverse-event? AND (Disproportionality OR “reporting odds ratio” OR “ROR” OR “proportional reporting ratio” OR “PRR” OR “information component” OR “Empirical Bayes geometric mean” OR “EBGM”).

#### 2.1.1. Inclusion Criteria

Adverse event study reporting on disproportionality statistics;Evaluating one named drug or a single class of drug;Comparing proportions of adverse event reports;Study aimed at generating signals for any adverse events with the drug;Disproportionality analysis conducted based on statistical parameters specified in previously published literature (for instance, there must be three or more cases of the event) [13,14,15];Analysis subsequently identified significant proportional increase in reports for one or more adverse events that the authors reported to be new or novel.

#### 2.1.2. Exclusion Criteria

Pre-specified or a priori hypothesis-testing study;Focused only on a specified single adverse effect, or adverse effects solely related to pre-specified organ systems;Vaccine studies.

### 2.2. Screening Studies for Inclusion

A clinical pharmacologist (YKL) with 25 years of experience in drug safety reviewed all titles and abstracts retrieved from the search (search hits: 414). Studies that met the inclusion criteria were entered onto a spreadsheet by one author (NN) and cross-checked against the full-test version by a clinical pharmacologist (YKL).

### 2.3. Data Extraction

We extracted information on the specific adverse events identified in the publication where the researchers indicated that there was significantly disproportionality (that indicated a potentially new ADR signal) in their analysis.

Follow-up measurements: For each specific adverse event that had been listed, we checked whether hypothesis-testing studies reporting on the statistical significance of the new ADR signal had been conducted.

This was based on two approaches which involved searching the following databases:PubMed using the adverse effect term and the name of the pharmacological compound or the class of the compounds;Google Scholar for citing articles related to the original proportionality analysis.

We aimed to extract, where available, information on the type of study, data source and statistical findings of any subsequent hypothesis-testing studies of the ADR signal.

## 3. Results

The number of hits from the initial PubMed search are shown according to year of publication in Figure 1.

We then narrowed down the search to focus on more recent articles published within the years 2017–2021.

After de-duplication, there were 414 articles on which screening of titles and abstracts was conducted.

Figure 2 shows the flowchart of study selection.

After applying the inclusion and exclusion criteria, we found 56 new suspected ADRs arising from 24 drugs (involving a wide range of antibiotics, biologics and CNS drugs) [16,17,18,19,20,21,22,23,24,25,26,27,28,29,30,31,32,33,34,35].

### 3.1. Literature That Cited the Study Where an ADR Signal Had Been Newly Identified

Forward citation tracking using Google Scholar showed that ADR studies had been cited a median of seven times (range 0–61). However, when we checked the content of the citing literature, none of the suspected new ADRs had undergone detailed evaluation or validation in the subsequent articles.

### 3.2. PubMed Search for Subsequent Evaluations of New ADR Signal

Similarly, we checked the retrieved articles from our PubMed search, but this did not reveal any confirmation studies for the 56 suspected new ADRs. We note that there was one ADR (spontaneous abortion with antivirals) where a subsequent cohort study reported that there was no association between the suspected drugs and adverse birth outcomes.

Table 2 gives full details of the drugs, new ADR signals as well as details of subsequent citing literature and any further detailed assessment of the ADR signal.

## 4. Discussion

Our study looked at recently published signals arising from spontaneous adverse event reporting systems in the years 2017 to 2021. We used a two-source search to determine whether these recent signals had been further clarified, particularly if they had been confirmed or refuted through further, more formal detailed studies.

Our findings regarding the lack of formal follow-up or methodologically rigorous evaluation means that readers of these published papers cannot be certain about the reliability of the reports in robustly identifying valid signals. Overall, we believe that our study lends confirmatory evidence to support Khouri’s assertion that most of the signals from disproportionality analyses are ‘unnoticed or unresolved’ [11].

The expanding number of disproportionality analyses conducted outside of regulatory authorities and pharmaceutical companies also raises interesting questions for traditional pharmacovigilance models.

The challenge of confirming true signals or false alarms is a particularly vexing problem because neither the regulator nor the pharmaceutical company should be ignoring potentially genuine signals (even if arising from unverified third-party sources), but they also clearly cannot be giving credibility to unrelated adverse events from less well-conducted and reported studies. This current profusion of signals from disproportionality analysis coupled with the absence of confirmation is a very difficult situation for regulators and manufacturers who are entrusted with communicating the safety of the named products.

There is a competing and, on the surface, reasonable argument that all safety signals should be taken seriously, ensuring that healthcare professionals and patients are fully informed, with regulators and pharmaceutical companies working together with academic researchers to assess risk from each significant signal. However, there are certain special methodological features of this type of analysis that lead us to be more cautious and far less certain regarding its reliability [2]. The first limitation is the lack of denominator data; we do not know the number of people receiving the drug, as we only have the number of suspected ADR reports. This means that we cannot judge the rate or incidence of the suspected ADR in a defined quantifiable set of users. We know nothing about the characteristics of the patients, their disease, comorbid conditions, and other potentially interacting medications. This means that the analysis is almost inevitably confounded by third factors that could have accounted for the differences in proportions of reported ADRs.

Other researchers have also identified potential reasons why false positives may occur with this type of analysis. Publicly available datasets are free to access, and no specific drug safety expertise is required [6]. The analytic method is basically carried out on 2 × 2 tables, and there are online calculators that can produce the results within seconds after keying in the relevant number of reports. It is therefore possible to conduct hundreds of comparisons in a relatively short time with few resources, and without the need to formulate and pre-specify a hypothesis with biological plausibility [6]. Hence, there is a major risk of false positives arising from multiple comparisons and significance testing [5]. This is accompanied by a serious risk of bias from selective outcome reporting, where multiple comparisons can be run until some significant findings are identified and judged suitable to be written up for a journal article. Here, we summarize the key attributes and limitations of analyses that rely on spontaneous adverse event reporting databases (Table 3).

We were not able to identify any similar studies to this current one. However, our previous research using similar methodology has reported on the follow-up and evaluation of 63 published case reports of suspected ADRs [12]. Overall, the majority (52/63, 83%) of these ADR reports had not undergone detailed evaluation, even at a time point of almost six years after initial publication. The citation tracking and database search found that there were only three occasions where hypothesized associations between drugs and adverse events were subsequently supported by controlled clinical studies. We believe that our current evaluation (based on published disproportionality analyses) has identified a similar lack of follow-up to that of the case reports review.

A number of other researchers have attempted to clarify the value of signal generation from spontaneous reports. Beau-Lejdstrom randomly selected 150 drugs from the United States database and looked for adverse effects where reporting odds ratios were available to be compared with corresponding matching adverse effects data from Cochrane systematic reviews [9]. This study evaluated a total of 125 adverse effects involving six drugs and found only a weak correlation between the reporting odds ratios from the spontaneous adverse events system and the corresponding odds ratios from the systematic reviews. Here, the authors concluded that the risk estimates from disproportionality analysis cannot be relied upon when making judgements on the actual risk of specific ADRs. Indeed, this study argues that there is no evidence to support the use of spontaneous adverse events databases for the purposes of making causal inferences about ADR signals. It appears that the presence of significant disproportionality does not on its own indicate a strong causal relationship between the drug and a suspected new ADR.

In contrast, Macia-Martinez et al. used a different approach to select ADRs for comparison of the disproportionality analyses to the relative risk estimates obtain from observational studies [10]. Unlike the aforementioned study above where ADRs were randomly selected, this particular evaluation was conducted on ADRs where there was already well-established evidence for European regulatory action. A total of 15 drug and adverse event pairs were chosen for evaluation. Here, the authors reported a significant correlation between the findings of the relative risk in observational studies and the results of the disproportionality analysis from the spontaneous report database. However, it is worth noting that a key weakness of this study is that the ADRs were preselected based on knowledge of an important relationship between the drug and adverse event. This could have potentially biased the results in favour of finding a relationship between the two analytical methods.

There are some limitations to our evaluation. We focused our search on suspected ADR signals from articles that have been published in PubMed in the last few years. This is because PubMed is a well-established and publicly accessible database, and therefore any reported signals have a relatively higher chance of being picked up and investigated by a wider audience than with proprietary databases that require fees for subscription. It is also possible that studies investigating the suspected ADR may not have cited the source that we identified. Equally, subsequent follow-up studies may be yet to be completed or they may not have been published in journals indexed in PubMed or Google Scholar. Even if this was the case, it does not detract from our findings that follow-up studies, if any do exist, are not readily accessible to clinicians and patients, thus meaning that the situation continues to remain uncertain in many people’s minds. Finally, we are aware that there are a multitude of different approaches towards disproportionality analysis, and that findings with greater validity may have arisen if we had selected adverse events with higher frequency such as ≥5 cases. Nevertheless, the lack of verification of signals remains a key finding of our study, even if we only considered specific adverse events where five or more cases had been analysed.

There is no doubt that spontaneous adverse event reporting systems have a crucial role to play in pharmacovigilance. Recent improvements include greater ease of access for members of the public and clinicians, thus promoting greater transparency in healthcare decision making. However, the data analysis and interpretation of spontaneous adverse event reports is extremely tricky, and may be prone to both false positives and false negatives. This is a particular concern when the disproportionality analysis of such spontaneous reports is conducted outside of a regulatory setting, with no subsequent detailed follow-up to either confirm or refute the signals that have been generated. From a methodology perspective, there needs to be far greater transparency and comprehensiveness in the reporting of such studies, whilst avoiding ‘spin’ [7,8]. It would be helpful to promote a wider appreciation of the major limitations of disproportionality analyses, as well as further development to overcome the substantial confounding (e.g., through matching classes of drugs with similar indications, age of patients, geographical sites of reports). We are aware of the importance of current initiatives to promote structured, comprehensive and transparent reporting for such studies [11].

## 5. Conclusions

Our study shows that the vast majority of signals arising from disproportionality analysis of spontaneous adverse events reporting systems have not been verified. We currently do not know whether such signals should be taken seriously or not. This creates a dilemma for healthcare professionals such as pharmacists who may come across published reports of new ADR signals but are not able to judge the validity of these reports when taking part in shared decision making regarding the safest treatment for their patients.

## Figures and Tables

**Figure 1 pharmacy-12-00033-f001:**
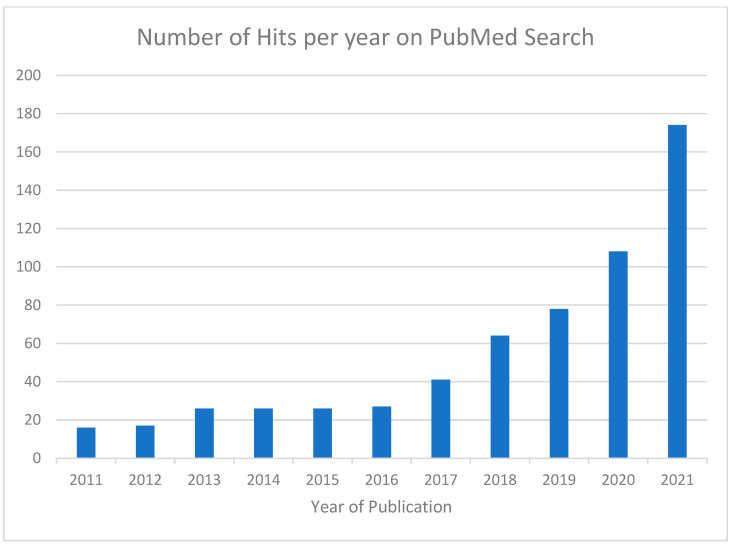
Number of hits from initial Pubmed search, demonstrating rapidly increasing numbers of articles reporting on disproportionality analysis. We then proceeded to select articles from 2017 to 2021 for further evaluation.

**Figure 2 pharmacy-12-00033-f002:**
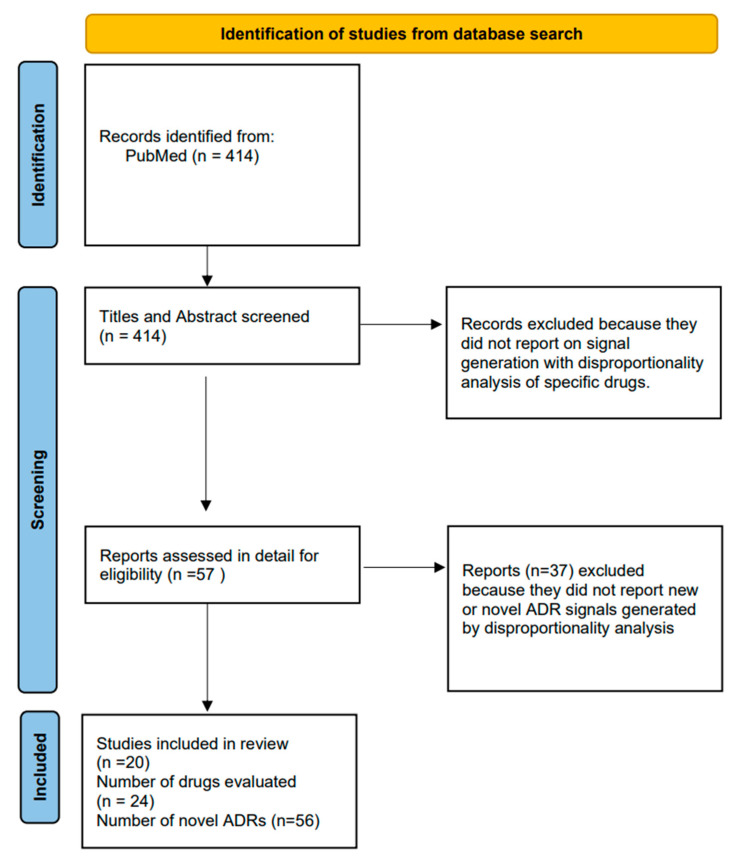
Flowchart of study selection.

**Table 1 pharmacy-12-00033-t001:** Important considerations for disproportionality analysis of adverse event reports.

Key features	Allows large-scale statistical evaluations of spontaneous reports (which might not otherwise be possible if the assessment required detailed individual examination of hundreds or even thousands of cases) Analysis can be programmed with pre-defined steps and parameters, thus speeding up processes through automation A variety of different statistical approaches are available Analysis can take place even if adverse events are incompletely reported and do not have full clinical data
Limitations	No clear consensus on optimal statistical approach Cannot estimate incidence of adverse events due to absence of denominator data Risk of bias from variation in reporting rates and confounding differences in patient characteristics and medication use.

**Table 2 pharmacy-12-00033-t002:** Drug compounds associated with new suspected ADR signals and any associated confirmation studies of the signals.

Study ID	Number of Drugs with Signal	Compounds and Comparators	Number of New ADR Signals	Specific Significant New ADRs	Times Cited (Google Scholar)	Confirmation Studies (Google Scholar)	Confirmation Studies on Pubmed Search
Choi 2020 [16]	1	Topiramate vs. other antiepileptics	1	steatorrhoea	4	0	0
Choi 2021 [17]	1	Cefatrizine vs. other antibacterials	2	corneal oedema, corneal ulceration	5	0	0
Cross 2019 [18]	1	Vedolizumab vs. TNF antagonists	2	CNS haemorrhages and stroke	15	0	0
Gastaldon 2021 [19]	1	Esketamine vs. all other drugs	1	suicidal ideation	61	0	No confirmation of such a link in research conducted prior to or after this paper.
Gatti 2021 [22]	1	Tedizolid vs. linezolid	1	hepatic failure	7	0	0
Gatti 2021 [21]	1	Tocilizumab vs. all other drugs	1	acute pancreatitis	42	0	0
Gatti 2021 [20]	1	Sacubitril/ valsartan vs. other cardiovascular drugs	4	sudden cardiac death, nipple pain, hepatic cyst, pyoderma gangrenosum	21	0	At least 3 studies looking at efficacy at reducing sudden cardiac death, not as ADR. No studies on other ADRs.
Gatti 2021 [23]	2	Ceftolozane tazobactam, ceftazidime avibactam vs. all other drugs	3	agranulocytosis, pancytopaenia, acute pancreatitis	7	0	0
Ha 2020 [24]	1	Infliximab vs. all other drugs	2	palpitation, temperature sensation change	6	0	No confirmation studies for infliximab.
Heo 2021 [25]	1	Doxycycline vs. all other drugs	8	malaise, ileus, confusion, malignant neoplasm, ectopic pregnancy, ovarian hyperstimulation, vaginal haemorrhage, bone necrosis	0	0	0
Lee 2021 [26]	1	Drospirenone vs. other contraceptive pills	3	chest pain, dyspnoea, fatigue	2	0	0
Merrison 2020 [27]	1	Encorafenib vs. other agents in class	2	Guillain–Barre syndrome, seizures	14	0	0
Omar 2021 [28]	4	crizotinib, ceritinib, alectinib, brigatinib	2	pneumothorax, photosensitivity	13	0	0
Park 2017 [29]	1	Imipenem vs. all other drugs	5	cardiac arrest, cardiac failure, myocardial infarction, Parkinson’s syndrome, and prostate enlargement	15	0	0
Peng 2020 [30]	1	Baricitinib vs. all other drugs	3	pneumocystis pneumonia, nephritis	0	0	0
Seo 2020 [31]	1	Paliperidone vs. other atypical antipsychotics	7	seborrhoea, obesity, breast neoplasm, vaginitis, fibroid, gingivitis, intervertebral disc disorder	2	0	0
Subeesh 2017 [32]	1	Vortioxetine vs. all other drugs	2	weight loss, ketoacidosis	12	1 (but weight loss trial preceded the FAERS signal)	Weight loss not confirmed in several other studies.
Tian 2021 [33]	1	Darunavir vs. all other drugs	4	neuropathy, diplopia, ptosis, ophthalmoplegia	2	0	0
Yang 2021 [34]	1	Entecavir/adefovir vs. other antivirals	1	spontaneous abortion	4	0	Retrospective cohort study reported no association with adverse birth outcomes.
Zhou 2021 [35]	1	Canagliflozin vs. all other drugs	2	cellulitis, osteomyelitis	2	Pooled analysis 2 RCTs—no increase in cellulitis	Inconsistent data on osteomyelitis. Non-significant in one meta-analysis and one observational study, but significant in another observational study (all studies preceded the disproportionality analysis).

**Table 3 pharmacy-12-00033-t003:** Attributes and key considerations when interpreting spontaneous adverse events data.

Attribute	Key Considerations
Data quality	Conflicting data may occur because reports on the same case can be submitted by many different parties. The completeness and accuracy of the submitted information are variable, and sometimes not all the necessary information is provided in the report.
Causal relationship	Contents of submitted reports are based on reporter’s judgement and opinions. Cannot be certain that adverse event was definitely due to the drug.
Rates of occurrence	Duplicate reports can occur. No denominator data regarding number of users and therefore incidence rates cannot be calculated or compared. Number of submitted reports cannot be used to infer the true risk because many other external factors can affect reporting.
Factors that influence reporting	Media publicity, recently launched drug, striking features of adverse event influence reporting.
Types of adverse events recorded	Particularly useful for events with low background rates and which happen soon after using a new drug. Less helpful if event of interest has a high background rate, or has similar features to disease progression.

## Data Availability

All extracted data are presented in Table 2.

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
