# Peer review of "New Adverse Drug Reaction Signals from 2017 to 2021—Genuine Alerts or False Alarms?"

_pharmacy, 2024, doi:10.3390/pharmacy12010033_

Round 1

Reviewer 1 Report

Comments and Suggestions for Authors

·         Interesting but provides no real new data and ignores the role of the report that are filed with regulatory agencies. The number of these reports would be substantially  higher than those that are published in various medical journals.  It also ignores what type of changes were made in the prescribing information over the same time-frame related to the spontaneous reports to the regulatory agencies. 

·         The report also did not evaluate whether the published literature used a Naranjo evaluation to determine the likelihood of the adverse reaction relative to the initiation of the drug therapy.

·         Idea behind the article is interesting and has some merit. However, the article is not written well and doesn’t appear to contain any relevant data. I read the paper a few different times and came away wondering what exactly the authors were trying to accomplish every time.

·         The main takeaway seems to be that the diversity and range of adverse reactions poses a major analytical challenge in that it’s too difficult to pinpoint an exact cause for them without knowing the patient’s entire medical history, medication regimen or specific data. I think this takeaway is already common knowledge. Do we need this paper to tell us that? I don’t think the takeaway from the article is anything new or innovative.

·         Was this supposed to be a systematic review? I’m assuming that was the intent based on the PRISMA diagram.

·         I would recommend writing inclusion/exclusion criteria in paragraph form, not using bullet points.

·         Lines 220-223. “First and foremost, clinicians and patients who are already using or are considering the use of drugs with these suspected new adverse reactions have to make up their minds regarding the optimal cause of action despite the absence of any confirmatory data.” This is already done in healthcare. This is common in healthcare.

·         Lines 225-231. “How should a pharmacist counsel the patient regarding risk of an unproven adverse event? Should additional monitoring for such events be put in place even with no clear basis? Would a patient opt to stop the current efficacious treatment because of the suspected adverse reaction?” This is not a new concept. Healthcare professionals have to do that all the time as many instances in healthcare aren’t “black and white.”

·         Line 246. “…With regulators and pharmaceutical companies rigorously investigating each occurrence.” May be plausible but may “clutter” any relevant data and introduce nonrelated adverse events that aren’t associated with a drug and due to another cause.

·         Line 261-262. “Without the need to formulate and pre-specify a hypothesis with biological plausibility.” Formulating and specifying a hypothesis gives purpose to the project and a paper. There is always a need for one.

If published:

·         Suggests the paper would be better as an editorial and already reads more like an editorial.

·         Introduction and discussion sections do not flow well. Filled with unnecessary examples that don’t add anything to the paper.

·         Paper does not read like an academic paper. Difficult to read with poor and opinionated, casual phrasing. Use of question marks throughout discussion is unnecessary and does not indicate professional writing. Paper would be improved with scientifically written, academic phrasing.

·         Good paper with decent ideas. Hard to understand what the authors were trying to prove/accomplish. Purpose of the article remains unclear.

·         Appreciated bullet point lists of inclusion/exclusion criteria but acknowledges this is rarely how this criterion is described in scientific papers.

·         Examining adverse events by drug class and displaying this in Table 1 instead may improve the paper. As the paper stands now, Table 1 seems to describe an entire variety of drugs and makes any conclusions about adverse events hard to reach.

General Comments

Methodology is questionable. Not clean or clear-cut.

·         Same person searched/located all articles and abstracts AND cross-checked spreadsheet against full-text version. Another author entered studies meeting inclusion criteria into the spreadsheet. So only two people were responsible for gathering/screening studies?

·         Section 2.1 describes a constructed cohort of published reports from 2017-2021. Yet Figure 1 describes hits from 2011-2016.

·         Only used PubMed and Google Scholar. Limited resources overall utilized.

What conclusion should be made from Figure 1 or Table 1?

·         Unclear what this figure or table describes or accomplishes. Should be deleted.  

·         Why do we care about the number of hits from a PubMed search? What does this add to the paper?

·         Table 1 describes “times cited in Google Scholar.” What does this tell us? Why do we care? This tells us nothing about where an adverse event came from or context of the citation. Also shows 0 confirmation studies for majority of the studies so what ultimately were the findings?

Phrasing issues.

·         Lines 6-7, 30, 32, 43, 72-73.

·         Lines 177-181 & Lines 185-187 appear to be duplications and express the same idea.

·         The use of the word “capture” is very distracting. Better word choice necessary.

·         Lines 96-97 would be better described with a visual or in a supplement. We have no idea how to comprehend what they’re attempting to describe. Even so, the example doesn’t accomplish anything or help us comprehend the paper better.

·         Detected self-citation by authors. Lines 123-124 and Line 272.

·         Potential error in Line 176. “After de-duplication, there were 441 articles where screening of titles and abstracts were conducted.” Line 149 describes a number of 414 search hits and this is also displayed in the PRISMA diagram.

·         Line 157. “Hypothesis-testing of the statistical significance of the new ADR signal…” Unclear what this even means?

Comments on the Quality of English Language

It more an issue of flow and presentation and not the use of the English language. 

Reviewer 2 Report

Comments and Suggestions for Authors

The manuscript is of an important topic, well written and based on good quality research.  The manuscript is easy to read and well structure.  

The issue of adverse effects is well discussed and handled in the manuscript, but more definition of adverse events reporting systems is needed.  This is crucial for the reader to understand the manuscript.  What for example is meant with "spontaneous" reporting.  What are the effects of reporting.  What kind of reporting and research do medicine manufacturing companies and authorities do.  All this needs some explanation, a few pages, to be sure that the reader understands the context.

Reviewer 3 Report

Comments and Suggestions for Authors

In the present study, the authors determined whether new ADR signals reported in scientific literature had been followed up. In this respect, they used two academic databases: PubMed and Google Scholar. Please find below my observations:

-        Introduction – references are necessary for Lines 23-95

-        Reference section should be set according to Mdpi requirements.

-        In a new table, authors should present the drugs/drug classes used for disproportionality analysis.

-        Authors could summarize in a figure or table the advantages and limitations of disproportionality analysis

-         My main concern is represented by the number of signals presented in Table 1.  Despite the EMA recommendations (“a signal of disproportionate reporting is assumed when a case count of ≥5 in EV), the present study shows that the number of new signals was < 5, for most drugs. In my opinion, this is an important limitation of the study and must be discussed.

Some references should be useful:

https://www.ema.europa.eu/en/documents/other/screening-adverse-reactions-eudravigilance_en.pdf

https://www.mdpi.com/1424-8247/15/12/1536#B62-pharmaceuticals-15-01536

Round 2

Reviewer 1 Report

Comments and Suggestions for Authors

Thank you for making the revisions to the previous manuscript, these definitely improved the readability and usefulness of your manuscript.

In the abstract and method sections, you talked about search the PubMed to identify published studies in 2017-2021, but Figure 1 has data from 2011 to 2021. 

The methods, figures, results, and discussion need to address the reason for these different dates if they are going to be included in the manuscript and figure. 

.  

Author Response

"In the abstract and method sections, you talked about search the PubMed to identify published studies in 2017-2021, but Figure 1 has data from 2011 to 2021. "

We apologise for this error, and we have added a further sentence to the Figure 1 legend to explain that "We then proceeded to select articles from 2017-2021 for further evaluation."

Reviewer 3 Report

Comments and Suggestions for Authors

Author has satisfied all the queries. In my opinion the manuscript can be accept for publishing. 

Author Response

Thank you.